# COVID-19's psychological toll on oral health: A cross-sectional study in Iranian adults

**Mahsa Karimi**[1,2], **Mohammad Reza Khami**[1,2], **Shabnam Varmazyari** [ID][2,3] *, **Ahmad Reza Shamshiri**[1,2], **Mahmoud Hormozi**[4], **Nourhan M. Aly**[5], **Morẹ́nikẹ́ Oluwátóyìn Foláyan**[2,6,7]

1 Research Center for Caries Prevention, Dentistry Research Institute, Tehran University of Medical Sciences, Tehran, Iran, 2 Department of Community Oral Health, School of Dentistry, Tehran University of Medical Sciences, Tehran, Iran, 3 Dental Students' Scientific Research Center, School of Dentistry, Tehran University of Medical Sciences, Tehran, Iran, 4 Department of Psychology, Payame Noor University, Tehran, Iran, 5 Department of Pediatric Dentistry and Dental Public Health, Faculty of Dentistry, Alexandria University, Alexandria, Egypt, 6 Oral Health Initiative, Nigerian Institute of Medical Research, Yaba, Lagos, Lagos State, Nigeria, 7 Department of Child Dental Health, Obafemi Awolowo University, Ile-Ife, Nigeria

* shabnam.varmazyari@gmail.com

## Abstract

### Background

The Coronavirus disease 2019 pandemic increased global psychological distress, emotional distress, and sleep disturbances, all known risk factors for compromised oral health. Despite this, there is limited understanding of the impacts of these psychological factors on oral health in certain populations, including Iranians. Thus, the present study investigates the associations between sociodemographic characteristics, emotional distress, sleep pattern changes, tooth brushing frequency, and oral ulcer reports in a sample of Iranian adults during the Coronavirus disease 2019 pandemic.

### Materials and methods

This cross-sectional, web-based study collected data from Iranian adults between July and September 2022 using respondent-driven sampling. The Mental Health and Wellness questionnaire was used to gather information on sociodemographic characteristics, emotional distress, sleep pattern changes, toothbrushing frequency, and oral ulcer reports. Simple and multiple logistic regression served for statistical analysis.

### Results

Among the 240 participants, comprising 164 females and 76 males, with a mean age of 35.3 years (±13.3), 28 individuals (11.7%) reported reduced tooth brushing frequency, and 35 individuals (14.6%) reported oral ulcers. Male gender (OR = 2.75, p = 0.016) and sleep patterns changes (OR = 2.93, p = 0.01) increased the likelihood of reduced tooth brushing frequency. Additionally, being younger than 30 (OR = 2.87, p = 0.025) and fearing coronavirus transmission (OR = 3.42, p = 0.009) increased the odds of oral ulcers.

**Data Availability Statement:** Data cannot be shared publicly because of TUMS data sharing policies. Data are available from the TUMS Ethics

Committee (contact via Ethics@sina.tums.ac.ir and +9881633626) for researchers who meet the criteria for access to confidential data.

**Funding:** This research was supported by Tehran University of Medical Sciences (TUMS), grant number: 1400-2-133-54316.

**Competing interests:** The authors have declared that no competing interests exist.

## Conclusions

Male gender and sleep pattern changes were risk factors for reduced tooth brushing frequency among the present sample of Iranian adults during the Coronavirus disease 2019 pandemic. Additionally, being under 30 and fearing coronavirus transmission were identified as risk factors for oral ulcers in this population. To preserve and promote adults' oral health during public health crises, targeted educational initiatives, public health awareness campaigns, and integrated mental and oral healthcare approaches are encouraged.

## Introduction

The Coronavirus disease 2019 (COVID-19) pandemic affected millions of people across the globe [1]. It caused widespread disruptions in daily life and massively altered all aspects of individuals' health and well-being [1]. The loneliness and uncertainty brought about by the pandemic resulted in increased psychological distress, including rises in stress, anxiety, depression, suicidal behavior, and post-traumatic stress disorder [2, 3]. Emotional distress intensified, marked by increases in negative emotions such as anger, fear, frustration, and boredom, and declines in positive emotions [3]. Sleep quality also deteriorated, evidenced by frequent reports of insomnia, delayed bedtimes, daytime sleepiness, and nightmares [4, 5].

Psychological and emotional distress, along with sleep disturbances, are particularly detrimental to oral health. Two ways through which these conditions compromise oral health include neglect of oral hygiene habits and development of oral ulcers [6–9]. Their resulting neglect of oral hygiene is driven by a combination of factors, including diminished motivation and interest, reduced energy and enthusiasm, adoption of unhealthy behaviors and lifestyle choices, and disruption of routines and self-care activities [6, 7, 9–12]. Additionally, their subsequent oral ulcers arise from changes in immune cell numbers and functions, as well as elevated oxidative stress and inflammatory markers in both saliva and serum [8, 13–16].

Neglected oral hygiene and the presence of oral ulcers can lead to pain, infection, aesthetic concerns, hindered nutrition, and disrupted speech [17]. Beyond these immediate effects, their negative impacts on oral health can harm socialization, damage self-esteem, undermine systemic health, reduce overall quality of life, and impose significant financial challenges, especially in lower-middle-income countries [18–20].

Considerable knowledge gaps persist regarding the psychological impacts of the COVID-19 pandemic on oral health. While conflicting reports exist regarding increases, decreases, or no changes in oral hygiene habits during the pandemic, limited reporting is available on the underlying psychological reasons for such changes [21]. Additionally, while during the COVID-19 pandemic, oral ulcers commonly appeared as aphthous lesions, herpes zoster, herpetic gingivostomatitis, candidiasis, and reactivation of herpes simplex virus (HSV-1) [22, 23], their potential psychological and emotional contributors remain relatively understudied, with their occurrences primarily attributed to viral disruption of oral epithelial cells, chemotaxis of lymphocytes and neutrophils, and treatment side-effects [22]. Most studies have focused on the pandemic's impact on emotional distress, sleep disturbances, oral hygiene habits, and oral ulcers individually, rather than exploring these factors' interrelationships in this context [24–26]. Finally, the limited studies that explored these relationships did not include the Iranian population [24, 27–29]. This research gap is notable, given the significant psychological distress and sleep disturbances experienced by the Iranian population during the pandemic [30,

31] coupled with this population's vulnerability to oral health threats due to Iran's lower-middle-income status, poor baseline oral health conditions, limited dental insurance coverage, and dental care accounting for 15.5% of total household healthcare expenditure [17, 20, 32]. Thus, understanding the psychological impact of the COVID-19 pandemic on Iranians' oral health could mitigate both immediate and long-term post-pandemic oral health challenges and inform preparations for future public health crises.

Therefore, the present study evaluates the associations between the independent variables of sociodemographic characteristics, pandemic-induced emotional distress, and sleep pattern changes, and the outcome variables of tooth-brushing frequency and oral ulcer reports in a sample of Iranian adults during the COVID-19 pandemic. The study revealed significant associations between male gender, changes in participants' disrupted sleep patterns, and reduced tooth-brushing frequency, as well as between being aged under 30, fearing coronavirus transmission (a form of pandemic-induced emotional distress), and self-reporting oral ulcers.

## Methods

This cross-sectional, web based-survey study was conducted in Iran between July 1st to September 30th, 2022 following approval from ethics committee of Tehran University of Medical Sciences on August 29th, 2021 (ethics code: IR.TUMS.DENTISTRY.REC.1400.110).

### Sample size

Using the sample-to-item ratio guide of 5-to-1 [33–35] the minimum pre-survey sample size required for this study was determined to be 135 valid responses for its 27 independent study variables. This sample size would allow for conducting regression tests with up to eight predictors, maintaining a minimum probability level (p-value) of 0.05.

### Participants

Adults were considered eligible for participation if they were 18 and above, were able to understand the survey language, and had access to the survey via an electronic device with internet connection. Respondent-driven sampling was employed for recruitment, leading to a sample of 240 adults. The Iranian online platform, Porsline, was used to create the survey link that got shared on social media platforms such as Telegram, WhatsApp, and Instagram. Additionally, the link was sent to a convenience sample of cellphone numbers using the Short Message Service.

### Research tool and procedure

The questionnaire included an introductory section explaining study aims, assuring respondents of data confidentiality and anonymity preservation, and emphasizing the voluntary nature of participation. It also included the investigator's contact information. Respondents only proceeded to study questions after providing written informed consent. The questionnaire platform allowed one submission per participant, and submission was only possible after survey completion.

Data were collected using the Mental Health and Wellness (MEHEWE) questionnaire, an instrument that covers various aspects of mental health and wellbeing of adults in relation to the multidimensional impacts of the COVID-19 pandemic [36]. It was validated for global use with an overall content validity index of 0.83 [24].

The questionnaire was translated from English to Persian, and then back-translated to English, to ensure translation accuracy. The overall Cronbach's alpha for the Persian version

of the questionnaire was 70%. Necessary changes were made to bring this version closer to the English one. The finalized Persian questionnaire took on average 10–15 minutes to complete.

### Independent variables

Socio-demographic characteristics: Age (in years), gender (male, female, others), education level (none, primary school or lower, high school diploma, undergraduate education, post-graduate education), marital status (single, married, divorced/ separated, widowed), testing positive for COVID-19 (yes, no), and employment in healthcare (yes, no).

Pandemic-induced emotional distress: Respondents were asked about experiencing any of the following ten forms of emotional distress during the pandemic: fear of contracting COVID-19, fear of transmitting COVID-19, being worried about others, being others' target of stigma or discrimination (getting treated differently by others due to one's identity, demonstrating COVID-19 symptoms, or other relevant reasons), frustration or boredom, anxiety, depression, loneliness, anger, grief or a feeling of loss. Checking the box for each distress was categorized as having experienced that distress during the pandemic.

Sleep Pattern Changes: Respondents were asked about experiencing changes in their sleep patterns (sleeping less, sleeping more, or other deviations from normal sleep) during the pandemic. Checking the box for each change was categorized as having experienced that change during the pandemic.

### Outcome variables

Toothbrushing frequency: Respondents were asked: Did your frequency of tooth brushing change during the pandemic? The response options were 'Yes, it increased'; 'Yes, it decreased'; or 'No changes'

Oral ulcers: Respondents were asked: Did you experience oral ulcers during the pandemic? The response options were: 'Yes'; or 'No'.

### Statistical analysis

We downloaded all the responses from Porsline onto a Microsoft Excel 2013 sheet. After encoding and organization, the data were transferred to IBM SPSS Statistics version 26 for Windows (IBM Corp., Armonk, N.Y., USA) for statistical analysis.

Descriptive statistics were calculated as frequencies and percentages for qualitative variables, and means, and standard deviations (SD) for quantitative variables. Since the outcome variables were categorical and dichotomous, simple and multiple models of logistic regression served to assess their associations with the independent variables. Independent and confounding variables with p-values less than 0.2 in simple logistic regression (Tables 2 and 3) were included in multiple logistic regression using the Forward method (Table 4). Sociodemographic variable categories of less than 20 frequencies were combined with a neighboring category with similar results in each model (S1 and S2 Files). Significance was set at the level of 5% for all tests.

### Results

Table 1 demonstrates that 240 individuals completed the questionnaire. The participants' mean age was 35.3 with a standard deviation (SD) of 13.3 and a range of 19 to 97. The study population consisted of 164 (68.3%) females, 125 (52.1%) married respondents, and 198 (82.5%) university-educated individuals. Out of the participants, 28 (11.7%) reported

**Table 1. Distribution of sociodemographic characteristics, reduced tooth brushing frequency, and oral ulcer reports among participants (n = 240).**

| Variable categories | N (%) | Reduced tooth-brushing frequency n (%) | Self-reported oral ulcer n (%) |
|---|---|---|---|
| **Gender** | | | |
| Male | 76 (31.7) | 14 (18.4) | 13 (17.1) |
| Female | 164 (68.3) | 14 (8.5) | 22 (13.4) |
| **Marital Status** | | | |
| Single | 107 (44.6) | 12 (11.2) | 19 (17.8) |
| Married | 125 (52.1) | 16 (12.8) | 14 (11.2) |
| Separated/Widowed | 8 (3.3) | 0 | 3 (40.0) |
| **Education Level** | | | |
| Primary school or lower | 3 (1.2) | 0 | 0 |
| High school diploma | 39 (16.3) | 4 (10.3) | 7 (17.9) |
| Undergraduate education | 123 (51.2) | 14 (11.4) | 19 (15.4) |
| Postgraduate education | 75 (31.3) | 10 (13.3) | 9 (12.0) |
| **COVID-19 status** | | | |
| Positive | 95 (39.6) | 14 (14.7) | 19 (20.0) |
| Negative | 145 (60.4) | 14 (9.7) | 16 (11.0) |
| **Employed in Healthcare** | | | |
| Yes | 100 (41.7) | 15 (15.0) | 15 (15.0) |
| No | 140 (58.3) | 13 (9.3) | 20 (14.3) |

reductions in their tooth brushing frequency and 35 (14.6%) reported experiencing oral ulcers during the COVID-19 pandemic.

Table 2 demonstrates that being male (OR = 2.47; 95% CI: 1.09 to 5.37), being the target of others' stigma or discrimination (OR = 3.33; 95% CI: 1.09 to 10.18), and changes in sleep patterns (OR = 2.59; 95% CI: 1.17 to 5.77) were related to reduced tooth brushing frequency during the COVID-19 pandemic.

**Table 2. Associations between sociodemographic characteristics, emotional distress, sleep pattern changes, and reduced tooth-brushing frequency of participants (n = 240).**

| Variable categories | Odds ratio | 95% CI | p-value |
|---|---|---|---|
| Age | | | |
| 29 or younger | Ref.[a] | - | - |
| 30–39 years-old | 1.48 | 0.59 to 3.68 | 0.40 |
| 40 or older | 0.87 | 0.27 to 2.81 | 0.82 |
| Gender | | | |
| Female | Ref. | - | - |
| Male | 2.42 | 1.09 to 5.37 | 0.03 |
| Marital status | | | |
| Married | 1.26 | 0.57 to 2.79 | 0.57 |
| Alone | Ref. | - | - |
| Education level | | | |
| Diploma and lower | Ref. | - | - |
| Undergraduate education | 1.46 | 0.43 to 4.98 | 0.54 |
| Postgraduate education | 1.22 | 0.38 to 3.94 | 0.74 |
| Positive COVID-19 test | | | |
| Yes | 1.62 | 0.73 to 3.57 | 0.23 |
| No | Ref. | - | - |

*(Continued)*

**Table 2.** (Continued)

| Variable categories | Odds ratio | 95% CI | p-value |
|---|---|---|---|
| Health-care worker | | | |
| Yes | 1.72 | 0.78 to 3.81 | 0.18 |
| No | Ref. | - | - |
| Fear of contracting COVID-19 | | | |
| Yes | 1.47 | 0.67 to 3.23 | 0.34 |
| No | Ref. | - | - |
| Fear of transmitting COVID-19 | | | |
| Yes | 0.92 | 0.41 to 2.06 | 0.84 |
| No | Ref. | - | - |
| Worrying about others | | | |
| Yes | 1.77 | 0.69 to 4.56 | 0.24 |
| No | Ref. | - | - |
| Stigma or discrimination | | | |
| Yes | 3.33 | 1.09 to 10.18 | 0.04 |
| No | Ref. | - | - |
| Frustration and boredom | | | |
| Yes | 1.65 | 0.62 to 4.42 | 0.31 |
| No | Ref. | - | - |
| Anxiety | | | |
| Yes | 0.95 | 0.42 to 2.21 | 0.9 |
| No | Ref. | - | - |
| Depression | | | |
| Yes | 1.76 | 0.74 to 4.15 | 0.2 |
| No | Ref. | - | - |
| Loneliness | | | |
| Yes | 1.89 | 0.77 to 4.62 | 0.16 |
| No | Ref. | - | - |
| Anger | | | |
| Yes | 1.75 | 0.76 to 4.04 | 0.19 |
| No | Ref. | - | - |
| Sadness and grief | | | |
| Yes | 1.71 | 0.74 to 3.94 | 0.21 |
| No | Ref. | - | - |
| Sleep pattern changes | | | |
| Yes | 2.59 | 1.17 to 5.77 | 0.02 |
| No | Ref. | - | - |

[a]Ref.: reference categories with which other categories of each variable were compared in the regression model.

Table 3 demonstrates that age (OR = 2.73; 95% CI: 1.10 to 6.75), fear of transmitting COVID-19 (OR = 3.36; 95% CI: 1.33 to 8.44), depression (OR = 2.36; 95% CI: 1.10 to 5.09), and sleep pattern changes (OR = 2.59; 95% CI: 1.17 to 5.7) were associated with higher odds of self-reported oral ulcers.

Table 4 demonstrates that being male (OR: 2.75, 95% CI: 1.21 to 6.25) and experiencing changes in sleep patterns (OR: 2.93, 95% CI: 1.29 to 6.66) led to higher odds of reduced tooth brushing frequency. Moreover, individuals younger than 30 had higher odds of self-reporting oral ulcers compared to those aged 40 and above (OR: 2.87; CI 95%: 1.14 to 7.19), and those

**Table 3. Associations between sociodemographic characteristics, emotional distress, sleep pattern changes, and oral ulcer reports of participants (n = 240).**

| Variable categories | Odds ratio | 95% CI | p-value |
|---|---|---|---|
| Age | | | |
| 29 or younger | Ref.[a] | - | - |
| 30–39 years-old | 2.73 | 1.10 to 6.75 | 0.03 |
| 40 or older | 1.23 | 0.39 to 3.98 | 0.72 |
| Gender | | | |
| Female | Ref. | - | - |
| Male | 1.33 | 0.63 to 2.81 | 0.45 |
| Marital status | | | |
| Married | 0.56 | 0.27 to 1.17 | 0.12 |
| Alone | Ref. | - | - |
| Education level | | | |
| Under diploma | Ref. | - | - |
| University degree | 0.68 | 0.23 to 1.99 | 0.48 |
| Higher education | 0.91 | 0.35 to 2.36 | 0.85 |
| Positive COVID-19 test | | | |
| Yes | 2.02 | 0.98 to 4.15 | 0.06 |
| No | Ref. | - | - |
| Health care worker | | | |
| Yes | 1.06 | 0.51 to 2.19 | 0.88 |
| No | Ref. | - | - |
| Fear of getting COVID-19 | | | |
| Yes | 1.39 | 0.68 to 2.85 | 0.37 |
| No | Ref. | - | - |
| Fear of transmitting COVID-19 | | | |
| Yes | 3.36 | 1.33 to 8.44 | 0.01 |
| No | Ref. | - | - |
| Worrying about others | | | |
| Yes | 1.99 | 0.83 to 4.78 | 0.13 |
| No | Ref. | - | - |
| Stigma or discrimination | | | |
| Yes | 1.76 | 0.54 to 5.70 | 0.35 |
| No | Ref. | - | - |
| Frustration and boredom | | | |
| Yes | 0.94 | 0.34 to 2.60 | 0.9 |
| No | Ref. | - | - |
| Anxiety | | | |
| Yes | 1.29 | 0.63 to 2.65 | 0.49 |
| No | Ref. | - | - |
| Depression | | | |
| Yes | 2.36 | 1.10 to 5.09 | 0.03 |
| No | Ref. | - | - |
| Loneliness | | | |
| Yes | 1.35 | 0.57 to 3.20 | 0.5 |
| No | Ref. | - | - |
| Anger | | | |
| Yes | 1.75 | 0.76 to 4.04 | 0.19 |

(*Continued*)

**Table 3.** (Continued)

| Variable categories | Odds ratio | 95% CI | p-value |
|---|---|---|---|
| No | Ref. | - | - |
| Sadness and grief | | | |
| Yes | 1.71 | 0.74 to 3.94 | 0.21 |
| No | Ref. | - | - |
| Sleep pattern changes | | | |
| Yes | 2.59 | 1.17 to 5.7 | 0.02 |
| No | Ref. | - | - |

[a]Ref.: reference categories that the other categories of each variable were compared to in the regression model.

with fears of transmitting the coronavirus had higher odds of self-reporting oral ulcers compared to others (OR: 3.42; 95% Cl:1.35 to 8.68).

## Discussion

To explore the impacts of the COVID-19 pandemic's psychological aftermath on Iranian adults' oral health, the present study investigated the relationships between sociodemographic characteristics, emotional distress, sleep pattern changes, reduced tooth brushing frequency, and self-reported oral ulcers in a sample of this population. It found that male gender and changed sleeping patterns increased the risk of reduced tooth brushing frequency. It also discovered that being under the age of 30 and fearing COVID-19 transmission heightened the risk of oral ulcer reports. Firstly, each of these findings is compared to the literature and explained. Then, all findings are jointly discussed for their practical implications.

Due to the lack of studies on gender-related oral hygiene differences in adults during the COVID-19 pandemic, comparisons were made with non-pandemic studies. Notably, recent studies in Saudi Arabian adults and an older study in the general Iranian population found that men had poorer tooth brushing habits than women [37–39]. These gender differences in tooth brushing have been attributed to men's lower oral health knowledge, more negative

**Table 4. Risk indicators for reduced tooth-brushing frequency and oral ulcer reports among participants (n = 240).**

| Outcome | Risk Indicators | Odds ratio | 95% Confidence Interval | p-value[a] |
|---|---|---|---|---|
| Reduced toothbrushing frequency | Gender | | | |
| | Female | Ref.[b] | - | - |
| | Male | 2.75 | 1.21 to 6.25 | 0.016 |
| | Changes in sleep patterns | | | |
| | Yes | 2.93 | 1.29 to 6.66 | 0.010 |
| | No | Ref. | - | - |
| Self-reported oral ulcers | Age | | | |
| | 29 or younger | 2.87 | 1.14 to 7.19 | 0.025 |
| | 30 to 39 | 1.38 | 0.43 to 4.43 | 0.584 |
| | 40 or older | Ref. | - | - |
| | Fear of transmitting COVID-19 | | | |
| | Yes | 3.42 | 1.35 to 8.68 | 0.009 |
| | No | Ref. | - | - |

[a]Only the significantly associated variables (p-value<0.05) produced by multiple logistic regression were reported in this table.
[b]Ref: reference category which other categories of a particular variable were compared within the regression model.

perceptions of dental care, higher self-evaluations of oral health, and lower compliance with care instructions [40, 41]. Men also visit the dentist less frequently than women, missing out on oral health education opportunities. This trend likely worsened during the pandemic, as dental care availability decreased both globally and in Iran [7, 42, 43]. Additionally, men tend to place less emphasis on esthetics and appearance, a factor noted in the general Iranian population and likely relevant to the male participants in the present study [39].

The relationship between changes in sleep patterns and increased odds of reduced tooth brushing frequency during the pandemic was similarly reported by a global study using the MEHEWE questionnaire which attributed this finding to sleep disturbances' association with decreased motivation and unhealthy behaviors [27]. Likewise, a pre-pandemic study of Indian dental students found that individuals with high-quality sleep flossed more regularly, attributing this regularity to lower levels of sleep-related psychological distress and fatigue [7]. These explanations align well with the previously noted associations between sleep disturbances and neglected oral hygiene through unhealthy lifestyle habits, disrupted routines, heightened psychological distress, and diminished motivation and energy [6, 7, 9, 11, 12]. In the present study, sleep disturbances' links with psychological distress may have been particularly influential, as the sample largely consisted of females, highly-educated individuals, and healthcare workers, all groups with comparatively greater psychological distress during the pandemic [44, 45].

In contrast to most of the research conducted both within and outside the context of the COVID-19 pandemic, the present study found that most oral ulcers reports came from adults aged under 30 [46–48]. Few studies have reported similar results, such as two pre-pandemic studies, one in South Africa identifying the highest prevalence among adults aged 25–34, and one in Iran indicating the highest prevalence among adults aged 30–40 [49, 50]. Younger adults frequently experienced greater psychological distress during the pandemic, as evidence by reports in the US, UK, and Australia, where adults under 30 faced higher psychological and emotional distress [51–53], where adults under 30 experienced higher psychological and emotional distress [51], negative emotionality [52], and post-lockdown mental distress [53]. Similar heightened psychological distress may have affected younger adults in the present study and contributed to their higher oral ulcers reports through altered immune cell function, increased cytokines, elevated cortisol and oxidative stress, and parafunctional habits like cheek and lip biting [15, 16].

The finding that fears of COVID-19 transmission increased the likelihood of reporting oral ulcers contradicted an Egyptian study that found no relationship between fears of COVID-19 contamination and experiencing oral ulcers [29]. However, it can find support in the global MEHEWE study [27], as well as studies in Nigeria [24], China [28], and Indonesia [54], all demonstrating links between pandemic-induced anxiety and stress and increased reports of oral ulcers. As the pandemic persisted, fears of coronavirus transmission likely evolved into anxiety and stress, contributing to oral ulcers by altering immune system function and increasing inflammatory markers [6, 10, 15, 16]. Such anxiety might have been particularly pronounced among the present study's participants, many of whom were healthcare workers facing heightened risks of direct exposure to the virus.

To safeguard adults' oral health amidst the ongoing psychological effects of the COVID-19 pandemic and prepare for future public health crises, targeted educational initiatives could offer practical tips for maintaining regular oral hygiene despite disruptions to daily routines. Gender-specific oral health education could also be integrated to address unique oral health needs. Additionally, public health awareness campaigns should focus on addressing emotional distress caused by public health crises, promoting effective coping strategies, highlighting the impact of psychological stress on oral health, and raising awareness about gender-specific

vulnerabilities during these times. Integrating mental health support with oral healthcare and vice versa could also be beneficial. It would allow professionals to screen at-risk individuals, proactively discuss concerns, provide tailored advice, encourage regular checkups, offer timely interventions, and make necessary referrals. Finally, strengthening collaboration between oral health professionals, mental health specialists, and public health authorities is essential. Such collaborations can lead to the creation of community outreach programs and the development of comprehensive care protocols that address both the oral health needs and psychological well-being of adults during public health crises.

### Strengths, limitations, and future directions

This study contributes to the literature by offering insight into the psychological impacts of the COVID-19 pandemic on a sample of Iranian adults' oral health, a field with limited research. It includes no missing data, contributes to the databank from the global MEHEWE study, and enables cross-border comparisons of pandemic's oral health impacts. However, the study also has a few limitations. The most important one is its small and skewed sample, predominantly comprising of females, university-educated individuals, and healthcare workers, which cannot be considered representative of the Iranian adult population. Several factors contributed to this outcome. Firstly, the respondent-driven sampling, chosen for efficiency and methodological alignment with the literature [24, 27], was a method that inherently limited representativeness and generalizability. Secondly, the web-based questionnaire, while improving efficiency, respondent anonymity [55], and safety per Iran's COVID-19 regulations, excluded individuals without internet access. Lastly, despite the extended data collection duration, the process was still hampered by Iran's frequent internet outages and social media bans in September 2022. The other study limitations include the potential for recall and social desirability biases, due to the use of self-report measures and the cross-sectional study design, though suitable for the study's aims, preventing the establishment of causality.

In conclusion and to address these limitations, we recommend the following measures: conducting qualitative studies to deeply explore the psychological impacts of the pandemic on oral health; employing longitudinal study designs to investigate the effects of integrated oral and mental healthcare provision on these impacts; using random sampling with larger sample sizes to enhance generalizability; incorporating offline and in-person clinical data collection methods to increase validity; and replicating findings across diverse populations and time frames.

### Conclusions

Despite extensive research on the relationship between oral and mental health, a substantial knowledge gap persists regarding their interplay during the COVID-19 pandemic, particularly in the Iranian population. The present study identified male gender and changes in sleep patterns as risk factors for reduced tooth brushing frequency during the pandemic. It also identified being under the age of 30 and fearing coronavirus transmission as risk factors for reporting oral ulcers during this period. Addressing these findings, the oral health of adults can be better supported in future public health crises through targeted educational initiatives, public health awareness campaigns, integrated mental and oral healthcare services, and collaborative mental and oral healthcare protocols.

### Supporting information

**S1 Dataset. Multiple regression for self-reported oral ulcers.**
(SAV)

**S1 File. Multiple regression for reduced tooth brushing frequency.**
(XLSX)

**S2 File. Minimum anonymized dataset.**
(XLSX)

## Acknowledgments

We extend our gratitude to all those who took part in this study.

## Author Contributions

**Conceptualization:** Mohammad Reza Khami, Mahmoud Hormozi, Nourhan M. Aly, Morẹ́nikẹ́ Oluwátóyìn Foláyan.

**Data curation:** Mahsa Karimi, Shabnam Varmazyari, Ahmad Reza Shamshiri.

**Formal analysis:** Mahsa Karimi, Mohammad Reza Khami, Shabnam Varmazyari, Ahmad Reza Shamshiri, Nourhan M. Aly, Morẹ́nikẹ́ Oluwátóyìn Foláyan.

**Funding acquisition:** Mahsa Karimi.

**Investigation:** Mahsa Karimi, Shabnam Varmazyari, Ahmad Reza Shamshiri, Mahmoud Hormozi, Nourhan M. Aly, Morẹ́nikẹ́ Oluwátóyìn Foláyan.

**Methodology:** Mohammad Reza Khami, Mahmoud Hormozi, Nourhan M. Aly, Morẹ́nikẹ́ Oluwátóyìn Foláyan.

**Project administration:** Mahsa Karimi, Mohammad Reza Khami, Shabnam Varmazyari, Ahmad Reza Shamshiri, Mahmoud Hormozi, Nourhan M. Aly, Morẹ́nikẹ́ Oluwátóyìn Foláyan.

**Resources:** Mahsa Karimi, Mohammad Reza Khami, Shabnam Varmazyari, Ahmad Reza Shamshiri, Mahmoud Hormozi, Nourhan M. Aly.

**Software:** Mahsa Karimi, Shabnam Varmazyari, Ahmad Reza Shamshiri.

**Supervision:** Mohammad Reza Khami, Morẹ́nikẹ́ Oluwátóyìn Foláyan.

**Validation:** Mohammad Reza Khami, Ahmad Reza Shamshiri, Mahmoud Hormozi, Nourhan M. Aly, Morẹ́nikẹ́ Oluwátóyìn Foláyan.

**Visualization:** Mahsa Karimi, Mohammad Reza Khami, Shabnam Varmazyari, Ahmad Reza Shamshiri.

**Writing – original draft:** Mahsa Karimi, Mohammad Reza Khami, Shabnam Varmazyari, Ahmad Reza Shamshiri, Mahmoud Hormozi, Nourhan M. Aly, Morẹ́nikẹ́ Oluwátóyìn Foláyan.

**Writing – review & editing:** Mahsa Karimi, Mohammad Reza Khami, Shabnam Varmazyari, Ahmad Reza Shamshiri, Mahmoud Hormozi, Nourhan M. Aly, Morẹ́nikẹ́ Oluwátóyìn Foláyan.

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
