## [Decision Letter · Decision Letter 0]

29 May 2024

PONE-D-24-17375Psychological toll of the COVID-19 pandemic on oral health: insights from a sample of Iranian adultsPLOS ONE

Dear Dr. Shabnam Varmazyari,

Thank you for submitting your manuscript to PLOS ONE. After careful consideration, we feel that it has merit but does not fully meet PLOS ONE’s publication criteria as it currently stands. Therefore, we invite you to submit a revised version of the manuscript that addresses the points raised during the review process.

We look forward to receiving your revised manuscript.

Kind regards,

Hadi Ghasemi

Academic Editor

PLOS ONE

Journal Requirements:

"This research was supported by Tehran University of Medical Sciences (TUMS), grant number: 1400-2-133-54316."

Reviewers' comments:

Reviewer's Responses to Questions

**Comments to the Author**

1. Is the manuscript technically sound, and do the data support the conclusions?

Reviewer #1: Yes

Reviewer #2: Yes

Reviewer #3: Yes

Reviewer #4: Partly

2. Has the statistical analysis been performed appropriately and rigorously? 

Reviewer #1: Yes

Reviewer #2: Yes

Reviewer #3: Yes

Reviewer #4: No

3. Have the authors made all data underlying the findings in their manuscript fully available?

Reviewer #1: Yes

Reviewer #2: Yes

Reviewer #3: Yes

Reviewer #4: Yes

4. Is the manuscript presented in an intelligible fashion and written in standard English?

Reviewer #1: Yes

Reviewer #2: Yes

Reviewer #3: Yes

Reviewer #4: Yes

5. Review Comments to the Author

Reviewer #1: Dear authors,

Thanks for sharing your work with us, the following points should be kept in consideration:

1. The manuscript within the scope of the journal.

2. Both the quality and data presentation of this manuscript are acceptable and of great importance to clinicians and even patients.

3. The manuscript expands our knowledge about COVID-19 and oral health .

4. The title should be revised and reduced its characters ( precise & and informative)

5. The abstract should reflect the content of the article and must be with range of 250-300 words.

6. Four to six keywords representing the main content of the article BUT not mentioned in the title.

7. More paragraphs should be incorporated to introduction about the details of COVID including complications. i.e. suggested references:

• Aldelaimi TN, Khalil AA, Alhamdani F. Herpes Zoster Post-COVID-19 Vaccine. Arab Board Medical Journal

2022;23:52-1.

• Aldelaimi A A, Aldelaimi T.N.. Mucormycosis in a Diabetic

Patient Post COVID-19. Journal of Clinical and Diagnostic Research. 2022 Mar, Vol-16(3): ZJ03, Journal of Clinical and Diagnostic Research. 2022 Mar, Vol-16(3): ZJ03 33. DOI: 10.7860/JCDR/2022/53108.16090

8. The statements in discussion are acceptable but few paragraphs about the justification of your findings and comparison with other recent relevant studies.

9. Up to date references should be kept in your reference list and the old should be omitted. i.e. Suggested reference:

Reviewer #2: Comments to the Author/s: -

I would like to begin by expressing my appreciation for the effort and dedication that the authors have put into this manuscript. My comments and suggestions to improve clarity and overall quality of the work are as follows.

Abstract

1. I prefer to remove the terms (independent variables) and (outcome variables).

2. I prefer not to use abbreviations in the abstract section, such as (MEHEWE).

3. Please type the mean and standard deviation as mean ± SD.

4. Keywords should not exceed five words.

Introduction

1. The reference in line 57 needs to be corrected.

2. Please clarify the following statement in line 66: "They might also initiate hormonal responses that weaken the immune system and thus, lead to the development of oral ulcers.

Materials

1. The date of approval needs to be written.

2. The required sample size is preferred to be written.

3. Authors sometimes use the term (sex) and at other times use the term (gender). Please use one of them consistently.

Discussion

1. The references in line 203 need to be corrected.

Reviewer #3: I respect the limitations of your research, but the importance of this topic, the size of your country, the population, and the different methods used to share it on social media platforms cannot be reflected in your small sample size. Therefore, this cannot be considered a representative Iranian sample.

Secondly, you use the term "mouth ulcer" in your outcome variables, which is considered jargon. It would be more appropriate to use "mouth lesion" instead.

Thirdly, you did not compare the negative and positive coronavirus subjects for outcome variables.

Finally, in the discussion on line 176, you stated that “being male and having changed sleeping patterns increased the chance of reduced tooth brushing frequency.” This is not accurate because the number of males in this study is much smaller than the number of females, so it does not accurately reflect the data.

Reviewer #4: 1-The survey has insufficient number of participants.

2-The discussion of this manuscript is insufficient.

3-The number of questions in the survey could have been more. Thus, it would have been a more comprehensive research.

4-There are no concerns about research ethics or publication ethics.

6. PLOS authors have the option to publish the peer review history of their article (what does this mean?). If published, this will include your full peer review and any attached files.

Reviewer #1: **Yes: **Tahrir Aldelaimi

Reviewer #2: No

Reviewer #3: No

Reviewer #4: No

---

## [Author Response · Author response to Decision Letter 0]

25 Jun 2024

Journal: PLOS ONE

Manuscript no: PONE-D-24-17375

Revised manuscript title: COVID-19's psychological toll on oral health: a cross-sectional study in Iranian adults

Date of revision: June 2024

The team of authors would like to thank the reviewers for their painstaking efforts, we found your comments extremely useful. Provided below is a point-by-point response to the reviewers’ comments. In addition, we conducted an extensive grammar check. All revisions were carried out with “track changes on”. 

In response to the journal’s comments: 

1. The manuscript was evaluated to ensure compliance with PLOS ONE style requirements, especially with regards to file names.

2. Amended funding statement was included in the cover letter.

3. The minimum anonymized dataset was attached as a supporting information file.

4. References were thoroughly checked, and no retracted articles are included to the best of our knowledge. 

Reviewer #1: 

Thanks for sharing your work with us, the following points should be kept in consideration: The manuscript within the scope of the journal. Both the quality and data presentation of this manuscript are acceptable and of great importance to clinicians and even patients. The manuscript expands our knowledge about COVID-19 and oral health.

RESPONSE: We thank the reviewer for this positive constructive feedback. 

4. The title should be revised and reduced its characters (precise & and informative)

RESPONSE: Title was revised to be more concise, precise, and informative in line with the requirement of the STROBE guidelines. It now reads as: “COVID-19's psychological toll on oral health: a cross-sectional study in Iranian adults”

5. The abstract should reflect the content of the article and must be with range of 250-300 words.

RESPONSE: All sections of the abstract especially the conclusion, were revised to ensure accurate reflection of the article. It is within the <300 word-limit range. 

6. Four to six keywords representing the main content of the article BUT not mentioned in the title.

RESPONSE: The keywords were limited to five representative ones that did not overlap with the title terms. 

7. More paragraphs should be incorporated to introduction about the details of COVID including complications. i.e. suggested references:

• Aldelaimi TN, Khalil AA, Alhamdani F. Herpes Zoster Post-COVID-19 Vaccine. Arab Board Medical Journal 2022;23:52-1.

• Aldelaimi A A, Aldelaimi T.N.. Mucormycosis in a Diabetic Patient Post COVID-19. Journal of Clinical and Diagnostic Research. 2022 Mar, Vol-16(3): ZJ03, Journal of Clinical and Diagnostic Research. 2022 Mar, Vol-16(3): ZJ03 33. DOI: 10.7860/JCDR/2022/53108.16090

RES RESPONSE: Additional paragraphs were incorporated into the introduction to elaborate on the effects of COVID-19 on oral health, particularly in terms of impacting oral hygiene behaviors and oral mucosa. The underlying mechanisms behind these impacts were also discussed in greater detail. While one of the suggested references was included, we regret that the other reference could not be included as it did not directly address COVID-19-related oral ulcers. Specific changes related to this comment can be found in lines 62-69 and 78-83. 

To maintain coherence and flow, the entire Introduction section underwent revisions for grammar and clarity. Furthermore, the paragraph outlining the research gap (lines 75-91) was expanded to align with the aforementioned expansions in the Introduction and provide a more thorough explanation of the identified gap.

8. The statements in discussion are acceptable but few paragraphs about the justification of your findings and comparison with other recent relevant studies.

RESPONSE: We tried to improve the robustness of the discussion by elaborating the explanations, comparisons, and justifications for each of the 4 main findings and also utilized more recent relevant studies. Since these changes resulted in notable expansion of Discussion, the practical implications of findings were summarized slightly and combined. You’ll find these changes in each paragraph of Discussion. As a result of changes to Discussion, slight changes also had to be made to Conclusion section and Abstract conclusion subsection. 

9. Up to date references should be kept in your reference list and the old should be omitted. i.e. Suggested reference:

RESPONSE: Newer references, particularly those published during and in the context of the pandemic, have been strongly prioritized. An effort was made to limit references to studies from after 2016 as much as possible and studies prior to this date were only included when absolutely necessary, for instance in the sample size determination subsection. 

Reviewer #2:

I would like to begin by expressing my appreciation for the effort and dedication that the authors have put into this manuscript. My comments and suggestions to improve clarity and overall quality of the work are as follows.

RESPONSE: We thank the reviewer for their constructive feedback. 

Abstract: I prefer to remove the terms (independent variables) and (outcome variables). I prefer not to use abbreviations in the abstract section, such as (MEHEWE). Please type the mean and standard deviation as mean ± SD. Keywords should not exceed five words.

RESPONSE: Outcome and independent variable terms were omitted from the abstract. The abbreviations MEHEWE and COVID were also omitted and replaced. The mean and standard deviation were referred to as mean ± SD. Five keywords were retained. 

Introduction:

The reference in line 57 needs to be corrected.

RESPONSE: Thanks for identifying this error. That reference is now corrected.

Please clarify the following statement in line 66: "They might also initiate hormonal responses that weaken the immune system and thus, lead to the development of oral ulcers.

RESPONSE: The sentence has been re-written in the process of the revisions outlined in response to reviewer 1. It now reads as: Additionally, their subsequent oral ulcers arise from changes in immune cell numbers and functions, as well as elevated oxidative stress and inflammatory markers in both saliva and serum.

Methods:

1. The date of approval needs to be written.

RESPONSE: Ethics approval date was included: on August 29th, 2021 ,…

2. The required sample size is preferred to be written.

RESPONSE: Thanks for raising this point. We included in the methods section: Using the sample-to-item ratio guide of 5-to-1 [33-35] the minimum pre-survey sample size required for this study was determined to be 135 valid responses for its 27 independent study variables. This sample size would allow for conducting regression tests with up to eight predictors, maintaining a minimum probability level (p-value) of 0.05.

3. Authors sometimes use the term (sex) and at other times use the term (gender). Please use one of them consistently.

RESPONSE: Thanks for raising this point. The term “gender” has now been used consistently throughout the paper. 

Discussion:

The references in line 203 need to be corrected.

RESPONSE: These references are now replaced due to the substantial revisions made to the discussion section per the requests of reviewers 1 and 4.

Reviewer #3: 

I respect the limitations of your research, but the importance of this topic, the size of your country, the population, and the different methods used to share it on social media platforms cannot be reflected in your small sample size. Therefore, this cannot be considered a representative Iranian sample

RESPONSE: We are grateful for your understanding. To address this comment, we included details about sample size determination in the Methods section and revised the Limitations’ subsection in Discussion comprehensively to outline the reasons for this shortcoming, explain the efforts made to mitigate it, and highlight the fact that present findings, although valuable, are not generalizable to the Iranian population. 

Secondly, you use the term "mouth ulcer" in your outcome variables, which is considered jargon. It would be more appropriate to use "mouth lesion" instead

RESPONSE: We thank the reviewer for suggesting. We opted for the term “oral ulcer” because it was used more conventionally in global studies such as the one conducted by Folayan MO, et al. (Hygiene. 2023; 3(2):85-92. https://doi.org/10.3390/hygiene3020009) and Folayan MO, et al. (Int J Environ Res Public Health. 2022 Sep 14;19(18):11550. doi: 10.3390/ijerph191811550). Thus, we hope the reviewer will agree with us that this is suitable terminology. 

Thirdly, you did not compare the negative and positive coronavirus subjects for outcome variables.

RESPONSE: The independent variable “positive COVID-19 test results” was included in descriptive analyses and simple logistic regression models, with their results demonstrated in Table 1,2, and 3. The variable was then included in the multiple logistic regression for oral ulcers since its simple logistics regression test produced a p-value of less than 0.2 (p-value=0.06) and excluded from multiple logistic regression for reduced tooth-brushing frequency since its simple logistic regression result did not meet the threshold for inclusion (p-value=0.23). After multiple logistic regression analysis, no significant association was spotted between the reports of oral ulcers and coronavirus test results and thus, this variable was not reported in Table 4 which includes only the variables that ended up being significantly associated with either reduced tooth brushing or oral ulcer reports following multiple logistic regression. The methods (statistical analysis) and Results (Table footnotes) sections were revised to accurately reflect these undertaken steps. 

Finally, in the discussion on line 176, you stated that “being male and having changed sleeping patterns increased the chance of reduced tooth brushing frequency.” This is not accurate because the number of males in this study is much smaller than the number of females, so it does not accurately reflect the data.

RESPONSE: Thank you for raising this point. Unequal sample sizes do not negate finding when simple and multiple logistic regression are used, as these tests automatically adjust for these imbalances through estimating the relationship between variables instead of counting and comparing raw sample counts. These tests then display these adjustments in their calculated p-values and confidence intervals. 

Reviewer #4:

1-The survey has insufficient number of participants.

RESPONSE: Thanks for this observation. We incorporated sample size estimation details into the Methods section and thoroughly addressed this shortcoming in the Discussion section's Limitations subsection by outlining its reasons, explaining the utilized mitigation strategies, and emphasizing the resulting negative implications for study representativeness and generalizability.

2-The discussion of this manuscript is insufficient.

RESPONSE: We tried to improve the robustness of the discussion by elaborating the explanations, comparisons, and justifications for each of the 4 main findings and also utilized more recent relevant studies. Since these changes resulted in notable expansion of Discussion, the practical implications of findings were summarized slightly and combined. You’ll find these changes in each paragraph of Discussion. As a result of changes to Discussion, slight changes also had to be made to Conclusions section and Abstract conclusion subsection. 

3-The number of questions in the survey could have been more. Thus, it would have been more comprehensive research.

RESPONSE: Your comment is valuable. We used a validated instrument for this survey that included the variables needed to address the study objectives originally inspired by the global MEHEWE study. However, the current findings can serve as hypotheses to conceptualize new studies by adding to the items of the questionnaire. 

4-There are no concerns about research ethics or publication ethics 

RESPONSE: Thanks for your valuable feedback.

---

## [Decision Letter · Decision Letter 1]

5 Jul 2024

COVID-19's psychological toll on oral health: a cross-sectional study in Iranian adults

PONE-D-24-17375R1

Dear Dr. Shabnam Varmazyari,

We’re pleased to inform you that your manuscript has been judged scientifically suitable for publication and will be formally accepted for publication once it meets all outstanding technical requirements.

Kind regards,

Hadi Ghasemi

Academic Editor

PLOS ONE

Additional Editor Comments (optional):

Reviewers' comments:

Reviewer's Responses to Questions

**Comments to the Author**

1. If the authors have adequately addressed your comments raised in a previous round of review and you feel that this manuscript is now acceptable for publication, you may indicate that here to bypass the “Comments to the Author” section, enter your conflict of interest statement in the “Confidential to Editor” section, and submit your "Accept" recommendation.

Reviewer #1: All comments have been addressed

Reviewer #2: All comments have been addressed

Reviewer #3: All comments have been addressed

2. Is the manuscript technically sound, and do the data support the conclusions?

Reviewer #1: Yes

Reviewer #2: Yes

Reviewer #3: Partly

3. Has the statistical analysis been performed appropriately and rigorously? 

Reviewer #1: Yes

Reviewer #2: Yes

Reviewer #3: Yes

4. Have the authors made all data underlying the findings in their manuscript fully available?

Reviewer #1: Yes

Reviewer #2: Yes

Reviewer #3: Yes

5. Is the manuscript presented in an intelligible fashion and written in standard English?

Reviewer #1: Yes

Reviewer #2: Yes

Reviewer #3: Yes

6. Review Comments to the Author

Reviewer #1: Thinks for sharing your work with us and thanks taking in consideration the suggested changes and comments

Reviewer #2: (No Response)

Reviewer #3: I want to thank the authors for their scholarly response and meticulous editing, which are sufficient to accept the manuscript for publication.

7. PLOS authors have the option to publish the peer review history of their article (what does this mean?). If published, this will include your full peer review and any attached files.

Reviewer #1: **Yes: **Tahrir N. Aldelaimi

Reviewer #2: **Yes: **Muhanad L. Alshami

Reviewer #3: No

---

## [Editor Report · Acceptance letter]

10 Jul 2024

PONE-D-24-17375R1 

PLOS ONE

Dear Dr. Varmazyari, 

I'm pleased to inform you that your manuscript has been deemed suitable for publication in PLOS ONE. Congratulations! Your manuscript is now being handed over to our production team.

Kind regards, 

on behalf of

Dr. Hadi Ghasemi 

Academic Editor

PLOS ONE